# Retrospective Evaluation of a Combination of Carboplatin and Bleomycin for the Treatment of Canine Carcinomas

**DOI:** 10.3390/ani12182340

**Published:** 2022-09-08

**Authors:** Antonio Giuliano, Angel Almendros

**Affiliations:** 1CityU Veterinary Medical Centre, City University of Hong Kong, Kowloon, Hong Kong; 2Department of Veterinary Clinical Sciences, Jockey Club College of Veterinary Medicine, City University of Hong Kong, Kowloon, Hong Kong

**Keywords:** carboplatin, bleomycin, canine carcinoma, urothelial carcinoma, nasal carcinoma

## Abstract

**Simple Summary:**

The safety of combined carboplatin and bleomycin chemotherapy treatment has never been assessed in dogs. Thirty dogs diagnosed with various types of carcinomas and treated with carboplatin and bleomycin chemotherapy were retrospectively evaluated. The treatment with carboplatin and bleomycin was well tolerated, with sixteen patients (53%) developing mild side effects. Gastrointestinal signs developed in thirteen (46%) of the dogs and hematological abnormalities in nine (30%). Objective response was observed in 24% of the cases (six partial responses) and 76% of cases achieved clinical benefit (partial response + stable disease). The combination of bleomycin and carboplatin appears safe and potentially effective for some types of carcinomas.

**Abstract:**

Carboplatin is a chemotherapy agent widely used in veterinary oncology to treat various types of tumors including carcinomas. Carboplatin has previously been used in combination with 5-Fluoro uracil (5-FU) or gemcitabine for the treatment of various carcinomas. Bleomycin is a chemotherapy drug commonly used in humans, but its use has been uncommonly reported in dogs. The combination of carboplatin and bleomycin chemotherapy treatment has never been reported in dogs. Dogs diagnosed with carcinoma and treated with a combination of carboplatin and bleomycin, at a single veterinary referral center, were retrospectively evaluated. Thirty patients met the inclusion criteria. The dose of carboplatin ranged from 200–250 mg/m^2^ (median 240 mg/m^2^) and the dose of bleomycin from 15–20 IU/m^2^ (median 15 IU/m^2^). The treatment with carboplatin and bleomycin was well tolerated, with sixteen patients (53%) developing side effects. Thirteen patients (46%) developed gastrointestinal signs and nine dogs (30%) developed hematological abnormalities. The most common side effects were grade-1 hyporexia and grade-1 neutropenia. Grade-2 neutropenia was rarely observed, and only one patient developed grade-3 neutropenia. None of the dogs developed grade-4 adverse events, or required hospitalization, or died due to the treatment. No signs of chronic side effects, including pulmonary toxicity, were observed. Objective response was observed in 24% of the cases (six partial responses) and 76% of cases achieved clinical benefit (partial response+ stable disease). Clinical signs improved in 24 of the 30 cases (80%). The main aim of this study was to evaluate the safety of bleomycin and carboplatin in combination for the treatment of various types of carcinomas. The combination of bleomycin and carboplatin appears safe and potentially effective for some types of carcinomas. Larger prospective studies are needed to confirm the safety and efficacy of combined carboplatin and bleomycin.

## 1. Introduction

Bleomycin is an antibiotic chemotherapy agent that has been used in human medicine for over 60 years [1]. Its mechanism of action is complex and partially unknown [2]. In the presence of iron and oxygen, activated bleomycin forms free radicals and causes double- and single-strand DNA breaks, leading to cancer cell apoptosis [3,4]. In human medicine, bleomycin has been effectively used in combination with various other chemotherapy drugs, including carboplatin [5,6]. Bleomycin in humans is used to treat different types of cancer, principally germline malignant tumors, Hodgkin’s lymphoma, and cutaneous and head and neck squamous cell carcinomas [5,6,7,8,9]. One of the advantages of using bleomycin as part of a multiagent chemotherapy protocol is its much lower myelotoxicity compared to other chemotherapy agents [10]. 

The most concerning reported side effects of bleomycin in human and animal models are pneumonitis and consequent pulmonary fibrosis [11,12]. Bleomycin hydrolase in the lungs detoxifies bleomycin, and its lack predisposes patients to pulmonary toxicity [13]. High cumulative bleomycin doses are an important risk factor for the development of lung fibrosis [11,14,15]. Pulmonary fibrosis has been reported rarely with cumulative adult doses below 450 IU [16]. More recently, renal impairment and cumulative doses higher than 300 IU have been reported to significantly increase the risk of pulmonary fibrosis in humans [10]. Bleomycin is not considered nephrotoxic at standard dosage, although dose reduction is necessary in patients with renal impairment [17] as it is mainly eliminated by the kidneys. Previous studies investigated bleomycin toxicity in experimental laboratory dogs, and found that frequent treatments with very high doses commonly caused gastrointestinal, dermatologic, renal, and severe pulmonary side effects [12,18].

Bleomycin has been infrequently used for the treatment of dogs with cancer. Bleomycin is considered to have modest single agent activity, especially due to its low cellular membrane permeability and low intracellular concentration [19,20,21]. Increased intracellular concentration can be obtained by electrochemotherapy and some studies reported significant clinical benefits and low side effects when bleomycin is given intralesional or intravenously in combination with electrochemotherapy [20,22,23,24]. Occasionally, bleomycin has also been used alone or in combination with cytarabine in relapsed canine lymphoma [25,26].

Carboplatin is a platin compound chemotherapy drug that is widely used in veterinary oncology. Carboplatin is used to treat various types of tumors including carcinomas and is considered standard care for adjuvant treatment of dogs with appendicular osteosarcomas [27,28,29].

Carboplatin is usually very well tolerated in dogs, but mild to moderate myelotoxicity and gastrointestinal adverse events are common. At a standard dose of 300 mg/m^2^, the risks of significant myelotoxicity are higher in dogs under 10 kg of weight [29,30,31]. Carboplatin is excreted almost entirely from the kidney, and dose adjustments in patients with renal impairment are usually needed [30,32].

The addition of bleomycin to carboplatin chemotherapy could be advantageous due to potential additive or synergistic effects without increased myelotoxicity. Bleomycin could potentially improve the efficacy of carboplatin, without appreciably increasing side effects. The coadministration of carboplatin and bleomycin in dogs with carcinoma has never been reported. 

The main aim of this study was to evaluate potential risks and side effects of a bleomycin and carboplatin chemotherapy protocol in dogs with various types of carcinomas.

## 2. Materials and Methods

Medical records from dogs treated with carboplatin and bleomycin for various types of epithelial malignant neoplasia were reviewed, from the clinical records of a single referral hospital (City University Veterinary Medical Centre) from December 2019 to October 2021, using the veterinary software management program RxWorks. The data were for all dogs that had a diagnosis of malignant epithelial tumor confirmed by a board-certified pathologist, via cytology or histopathology, and received concurrent administration of carboplatin and bleomycin. Data collected included age and weight at the time of diagnosis, the dog’s breed, the type and location of the carcinoma, stage of the disease, dose and number of chemotherapy treatments administered, side effects, and treatment response (when an objective measurable response could be assessed). All the patients in the study were treated in the same institution, resulting in a reasonably standardized treatment for each patient. Carboplatin and bleomycin were given a few minutes apart, both by slow intravenous (IV) boluses, every three weeks. Intravenous saline solution was administered a few minutes before and after the chemotherapy infusions to ensure patency of the IV catheter and the full administration of the drugs, avoiding extravasation and cytotoxic exposure accidents. The standard target doses for carboplatin and bleomycin were 200–250 mg/m^2^ (based on the size of the dog) and 15–20 IU/m^2^, respectively. Bleomycin in powder form was diluted in 5 mL of sterile saline for IV infusion, and carboplatin was given undiluted as an IV infusion. A complete blood count was performed in most cases at 7, 14 and 21 days, as well as before each chemotherapy treatment in all the cases. A biochemistry profile including renal parameters was performed before each treatment. More frequent or specific blood tests were performed as deemed necessary by the primary clinician. Dogs with significantly increased kidney parameters were not treated. Adverse events were graded based on the Veterinary Cooperative Oncology Group Common Terminology Criteria for Adverse Events (VCOG-CTCAE v2) [33]. Response rate was measured based on the Veterinary Cooperative Oncology group’s RECIST response in solid tumors [34]. Re-staging and response assessment were performed mostly every 3–9–12 weeks, however some cases were re-staged less or more frequently based on clinician decision and/or owners’ wishes. The response to treatment and subsequent monitoring were assessed by clinical examination, caliper measurement where appropriate, or by imaging, depending on the location of the tumor. Only patients with objectively measurable gross disease were assessed for response. Complete response (CR) was defined as resolution of all clinical and imaging-based evidence of disease; partial response (PR) was defined as a decrease of at least 30% in tumor diameter with no new lesions; stable disease (SD) was defined as a decrease of less than 30% or an increase of less than 20% in tumor diameter with no new lesions. Progressive disease (PD) was defined as an increase in tumor diameter greater than 20% or the development of new lesions. Overall response rate (ORR) was defined as CR + PR. Clinical benefit was defined as CR + PR + SD for at least six weeks. For all patients included in the study, the progression-free survival was calculated in days from the date of the first chemotherapy treatment to the date of disease progression. Survival time for each patient was calculated from the time the carcinoma was diagnosed to the time of death of the patient, including euthanasia. Median survival time and progression-free survival were calculated in days, with a Kaplan–Meier product-limit method. Patients that were alive, lost at follow up, or died of other diseases were censored. Statistics were performed with GraphPad Prism 9. Due to the small sample size and different heterogenous groups, statistics were only descriptive. 

## 3. Results

Thirty dogs met the inclusion criteria. Seventeen were male and thirteen females. The most common breed was the toy poodle (*n* = 8), followed by various other breeds, as summarized in Table 1. The median age of the dogs was 12 years old (mean 11.6 years old) and the median weight was 8 kg, mean 11.7 kg (range 2–45 kg). Eleven dogs had sino-nasal carcinoma, with six classified as stage III and five stage IV (Adam modified staging). Two of these sino-nasal carcinomas were squamous cell carcinomas, and one of the dogs with stage IV nasal carcinoma also had metastasis to the lungs. Another dog with stage IV nasal carcinoma also had disseminated epitheliotropic cutaneous and muco-cutaneous lymphoma. Three dogs had large, unresectable squamous cell carcinoma of the maxilla, and one lingual, all without distant metastasis. One dog had right tonsillar squamous cell carcinoma with metastasis to the right retropharyngeal and submandibular lymph nodes. One dog had a carcinoma of the left ear canal with extension to the surrounding tissues of the neck. One dog had a large cutaneous squamous cell carcinoma in the left antebrachium. Three had advanced lung carcinomas with intrapulmonary and trachea-bronchial lymph node metastasis. Five dogs had urinary bladder transitional cell carcinoma and three had prostatic carcinomas. Two of the three dogs with prostatic carcinomas had metastasis to the iliac lymph node. One dog had stage V mammary carcinoma with metastasis to the lungs. Data are summarized in Table 1.

Carboplatin dose ranged from 200 to 250 mg/xm^2^ (median 240 mg/xm^2^ and mean 231 mg/xm^2^). Doses were mainly chosen based on the weights of the dogs, with smaller dogs likely to receive lower dosage compared to larger dogs. Bleomycin was administered at a dose of 15–20 IU/xm^2^ (median 15 IU/xm^2^ and mean 15.4 IU/xm^2^). Patients were treated at three-week intervals and received a maximum of six cycles, ranging from one to six cycles with a median of three cycles. A maximum of six cycles was given to reduce costs and avoid risks of cumulative toxicity of bleomycin. All dogs received antiemetic premedication with a bolus of intravenous maropitant of 1 mg/kg, a few minutes before chemotherapy injection.

Most of the dogs (*n* = 25) were treated with carboplatin and bleomycin as a first line treatment. Of the other five dogs treated as a second line, one patient was previously treated with toceranib, one with metronomic chlorambucil, two with doxorubicin, and one with an investigational drug. All these patients progressed on the previous treatments, before being started on carboplatin and bleomycin. Of the patients that were started with second-line carboplatin and bleomycin, one dog with nasal carcinoma achieved stable disease, and one with a frontal sinus squamous cell carcinoma achieved partial response. One dog diagnosed with stage IV nasal carcinoma and epitheliotropic lymphoma was started on carboplatin and bleomycin as a first-line treatment. After the second cycle, due to progression of the epitheliotropic lymphoma, the carboplatin and bleomycin were alternated with lomustine, and later stopped due to lymphoma progression. This dog continued the treatment for her epitheliotropic lymphoma with various other chemotherapy drugs.

All of the dogs received meloxicam, except for three patients that were on prednisolone. Most of them received other analgesics and occasionally antibiotics when deemed necessary by the primary clinician.

Most of the dogs tolerated the chemotherapy protocol very well, with only occasional and mild side effects. In total sixteen dogs developed some form or combination of side effects (53%). Fourteen dogs (46%) developed gastrointestinal adverse events, nine dogs (30%) hematological abnormalities, and eight dogs (27%) a combination of gastrointestinal and hematological adverse events. Most of the gastrointestinal side effects were VCOG grade-1 hyporexia and/or vomiting, with only one grade-2 hyporexia. Most of the hematological abnormalities reported were grade-1 neutropenia and thrombocytopenia. Only one dog developed grade-3 neutropenia, and two dogs developed grade-2 neutropenia. Only one dog developed grade-1 thrombocytopenia, whereas two dogs developed grade-2 thrombocytopenia. The adverse events that were observed are summarized in Table 2.

The only dog that developed grade-3 neutropenia was not pyretic and was clinically well. This dog received the lowest dose of carboplatin, at 200 mg/m^2^, due to stage-1 chronic kidney disease (CKD) and a low body weight of 2.4 kg. The dose was reduced further to 180 mg/m^2^ and the treatment was continued every four weeks without any further complications. No progression of the renal disease occurred. One dog developed grade-1 followed by grade-2 progressive increase of urea and creatinine after the fourth and fifth treatment cycles. Meloxicam was stopped, and the dose of carboplatin was reduced and then stopped. The renal disease progressed despite stopping chemotherapy and meloxicam, and the dog was eventually euthanized due to renal failure 117 days later. No other dogs stopped the treatment due to side effects. One dog with pre-existing grade-2 increase of alanine transferase (ALT) and alkaline phosphatase (ALP), developed grade-3 ALT elevation, but this was considered unlikely to be related to the chemotherapy treatment. Meloxicam was stopped and the liver parameters improved, though remaining above the normal range. The chemotherapy treatment was continued without any delay or dose reduction; this dog died later due to concurrent cardiac disease. One of the dogs that developed grade-2 neutropenia also developed borderline high urea and creatinine with normal urine specific gravity (USG), but increased symmetric dimethylarginine (SDMA) at 26 ug/dL (reference interval 0–14 ug/dL). This dog continued the treatment with a carboplatin dose reduction (from 200 mg/m^2^ to 180 mg/m^2^), and the kidney parameters and SDMA remained stable. In total only three dogs required dose reduction and four dogs required dose delay. 

Most of the dogs that achieved an objective response completed the six cycles of chemotherapy treatment. However, one dog stopped after a single cycle due to financial concerns, and two dogs stopped after four cycles, one following the development of cardiac failure due to severe pre-existing mitral valve disease and the other due to the decision of the owner to switch to oral chemotherapy. The dog that developed renal failure stopped the treatment after five cycles. Clinical signs suspicious of lung toxicity were not seen in any cases and all the dogs that had radiography of the thorax taken as part of their re-staging showed no evidence of pulmonary changes. Other treatments including metronomic chemotherapy, toceranib, or both were started when chemotherapy was stopped due to progression of disease or when the animals completed six cycles of treatment. 

Objective response could be evaluated in 25 cases. Six dogs achieved partial response and thirteen dogs achieved stable disease, with 76% clinical benefit and 24% objective response. Clinical signs improved in 24 of the 30 cases (80%). One of the dogs that achieved partial response had a stage-4 nasal tumor with associated facial deformity and pulmonary metastasis visible on CT scan and radiography. After the first and second cycles of chemotherapy the facial deformity and exophthalmos resolved and the opacity in the lung was no longer visible by thoracic radiography. This dog remained free of disease for 310 days, at which the metastasis was again visible by radiography, but no clinical signs of recurrence of the nasal tumor were present. The dog was euthanized due to neurological signs, suspected to be related to the progression of the tumor 390 days later. The other dogs that achieved partial responses included one dog with stage IV nasal tumor and facial deformity, one dog with maxillary squamous cell carcinoma, one with a frontal sinus squamous cell carcinoma, one with a carcinoma of the ear canal, and one with a cutaneous squamous cell carcinoma. The dogs that achieved stable disease included two dogs with SCC of the maxilla, two with nasal carcinomas, seven with urothelial carcinomas (five of the bladder/urethra and two of the prostate), and two dogs with lung carcinomas. Of those with nasal carcinomas, only patients with facial deformity could be assessed clinically for objective responses, because only one repeated the CT scan. Interestingly in this case, the CT scan imaging repeated after four cycles of chemotherapy showed almost 20% reduction of the mass. Despite this mass reduction not being sufficient to be classified as a partial response, the dog achieved stable disease with complete resolution of clinical signs and lived for an additional 398 days. One of the dogs with stage-4 nasal carcinoma and facial deformity achieve completed resolution of the facial deformity, but a CR could not be confirmed due to the lack of verification by CT scan following treatment. This dog remained free of disease for 105 days but died due to congestive heart failure 130 days later.

The median progression free survival (PFS) of the dogs that achieved partial and stable disease (n. 19) was 310 days (range 45–666 days) and PFS for the all the dogs was 98 days (range 21–666 days). Median survival time (MST) for all the patients was 330 days (range 84–715 days); Figure 1. Median survival times for the two most numerous groups including dogs with sino-nasal carcinoma and urothelial carcinoma were 390 days and 608 days, respectively; Figure 2 and Figure 3. PFS and MST for each patient are reported in Table 1.

At the time of writing, five patients were still alive and five were lost to follow up. Although no post-mortem exams were performed, it was suggested that five dogs died of concurrent conditions, two of congestive heart failure due to pre-existing cardiac conditions, one of suspected GI perforation, one of renal failure, and one was euthanized due to progression of epitheliotropic lymphoma. One dog with maxillary SCC that achieved stable disease was reported as a sudden death. This dog was considered to have died from the pre-existing cancer, but other causes were not ruled out. All the remaining dogs died or were euthanized because of the diagnosed neoplasia.

## 4. Discussion

The main aim of this study was to retrospectively evaluate the safety of combining bleomycin and carboplatin in dogs with carcinoma. While not the primary aim, some objective and clinical responses to this protocol were noticed and appeared promising. 

The combination of carboplatin and bleomycin seemed well tolerated, and side effects were comparable to previous studies of patients treated with carboplatin alone [26]. 

Bleomycin is a chemotherapy drug used rarely in veterinary oncology. This could be due to the lack of experience from veterinary clinicians, lack of literature supporting its use, or lack of treatment efficacy such as in dogs with relapsed lymphoma [25]. Bleomycin is considered to have a low intracellular accumulation inside the cell, hindering is full clinical efficacy [20]. However, this drug has proven to be beneficial against a variety of cancers in humans and is considered a standard part of multimodal agent treatment in various human cancers [21].

Concerns about causing cumulative pulmonary fibrosis could also be another reason for limited use of bleomycin in dogs. In humans the development of pulmonary fibrosis is rare below 240 IU and 400 IU per adult, and at lower doses, only around 3–5% of people develop pulmonary fibrosis, with a reported 1–2% mortality rate in adults [35,36,37,38]. High and frequent dosage from 1.25–5 mg/kg every four days for 11 treatments or 0.65 mg/kg every four days for 31 treatments were reported to cause histopathologically detectable pulmonary fibrosis in dogs [18].

Bleomycin was used in this study for a maximum of six injections, three weeks apart. The cumulative doses in this group of patients ranged from 90 to 120 IU/m^2^. These were much lower doses than the cumulative doses reported to present risks for pulmonary fibrosis in laboratory beagles and in humans [18,35,36,37,38]. In this study, neither clinical signs nor thoracic radiography changes suggestive of pulmonary fibrosis were observed. However, more sensitive investigations for the detection of pulmonary fibrosis, including CT scan or lung biopsy, were not performed. Potential subclinical pulmonary fibrosis could remain a potential risk in some sensitive patients, especially in dogs with underlying renal disease. Carboplatin and bleomycin are both eliminated mostly unchanged in the kidney, and changes in kidney functionality are likely to increase the half-life of both drugs, potentially causing severe side effects. Renal parameters were monitored before each treatment, while glomerular filtration rate (GFR) was not monitored in any of the patients, as this monitoring is very rarely routinely performed in dogs. Assessment of early markers of kidney disease, such as SDMA, was only performed at the discretion of the clinician in suspicious cases of early CKD. Despite the lack of extensive kidney monitoring, most of the dogs did not develop any severe adverse effects and none of the dogs were hospitalized or died due to the treatment. 

The dog that died of renal failure was likely to have underlying subclinical chronic renal disease that manifested clinically during the treatment. This dog was also on meloxicam, and the renal disease progressed despite stopping all the drugs. Considering the CKD progressed slowly after stopping all the treatments, it is unlikely that the renal failure was triggered by the chemotherapy treatment. However, the possibility of potential insult and renal decompensation due to a combination of carboplatin, bleomycin, and meloxicam cannot be completely ruled out. No other dogs developed or died of renal disease; however, caution should be taken for patients with elevation of renal parameters, even if mild.

Hematological side effects were uncommon and very mild, with only one case developing grade-3 neutropenia and none of the patients needing hospitalization due to the treatment. However, only nineteen patients had weekly hematological assessments, and it is likely that dogs tested only at three-weekly intervals could have developed subclinical neutropenia or thrombocytopenia at nadir 12–15 days post carboplatin. Most of the dogs that started this novel protocol were screened weekly, however, as no significant side effects or abnormalities in the blood were noticed, to reduce cost of rechecks, the remaining patients were subsequently rechecked every three weeks, or sooner if there were clinical concerns.

Gastro-intestinal side effects were frequent but were mostly mild and self-limiting.

The aim of this study was primarily to assess the safety of combining carboplatin and bleomycin; however, twenty-four percent of dogs achieved a measurable objective response, showing some degree of efficacy for carboplatin and bleomycin chemotherapy in combination. It is difficult to assess, however, whether the objective responses could be attributed to the combination of carboplatin and bleomycin or solely to carboplatin.

Response to carboplatin alone has been reported in a few cases of oral and frontal sinus squamous cell carcinomas [39,40], and efficacy of carboplatin in combination with radiotherapy has also been reported in SCC in various locations [41,42,43]. Response to carboplatin and piroxicam has been reported in urothelial carcinoma [44,45]. Interestingly, in the only prospective study investigating response to carboplatin in urothelial carcinoma, 13% partial response and 54% stable disease were reported [44]. In the current study, no response was found in urothelial carcinoma, but the rate of patients achieving stable disease was similar. 

The treatment of various carcinomas with carboplatin in combination with other chemotherapies has been investigated in only two previous studies. In one study, carboplatin was administered in combination with fluorouracil, while in the other, carboplatin was added to gemcitabine. The response rates reported in these two studies were 43% and 13%, respectively [46,47]. In both studies, different proportions, types, and locations of carcinomas were included, so comparison is challenging. The side effects reported in previous studies combining carboplatin with 5-FU or gemcitabine were comparable to observations in the carboplatin and bleomycin protocol used in this study. However, lower myelotoxicity was found in the current study compared to the previous two, which reported a myelotoxicity of 58% and 56% when carboplatin was combined to gemcitabine and 5-FU, respectively. In both protocols occasional grade-4 toxicity was reported, and was not seen using the current protocol. Considering the similarity in dose of carboplatin used in previous and present studies, the reduced myelotoxicity in the current study is likely to be related to the very low myelotoxicity of bleomycin compared to other chemotherapy drugs. However, under-reporting of side effects, due to lack of weekly blood testing for every patient, was also possible.

There were some other limitations in this study that need to be taken into consideration. Most of the dogs that did not respond to chemotherapy were started on either metronomic chemotherapy and/or toceranib phosphate. This treatment approach could have increased significantly both PFI and MST. However, despite being outside the aim of this study, it is worth noticing that the two main groups, nasal and urothelial carcinomas, achieved long-term survival of 390 and 608 days respectively. The MSTs for nasal and urothelial carcinomas reported in the current study were higher compared to what has been reported in other previous studies [45,48]. It is possible that sequential treatment with carboplatin and bleomycin followed by toceranib and/or metronomic chemotherapy could be beneficial in these two types of carcinomas. However, due to the small number of cases for each group, the retrospective nature of the study, and the sequential treatment with various other drugs, the data on survival need to be considered cautiously and prospective studies with a larger number of cases for each group are necessary to evaluate this possibility. 

Most of the dogs that were treated with carboplatin and bleomycin also received a variety of supportive treatment including analgesics and the anti-inflammatory COX-2 inhibitor, meloxicam. As COX-2 inhibitors are considered effective in the treatment of some type of carcinomas, such as urothelial carcinoma, it is possible that the meloxicam treatment could have contributed to the response rate and improvement of clinical signs. The authors acknowledge some limitations in data recording and collection due to the retrospective nature of this study, as well as a lack of complete standardization for the monitoring and rechecks. 

## 5. Conclusions

In conclusion, despite all the limitations of a retrospective study, the combination of bleomycin and carboplatin appears safe and could be beneficial in some type of carcinomas. The use of a standard dose of carboplatin, especially in larger patients, and a higher dose of bleomycin could result in higher response rates while retaining an acceptable safety profile. However, a large prospective study is needed to assess the safety and efficacy of this drug combination.

## Figures and Tables

**Figure 1 animals-12-02340-f001:**
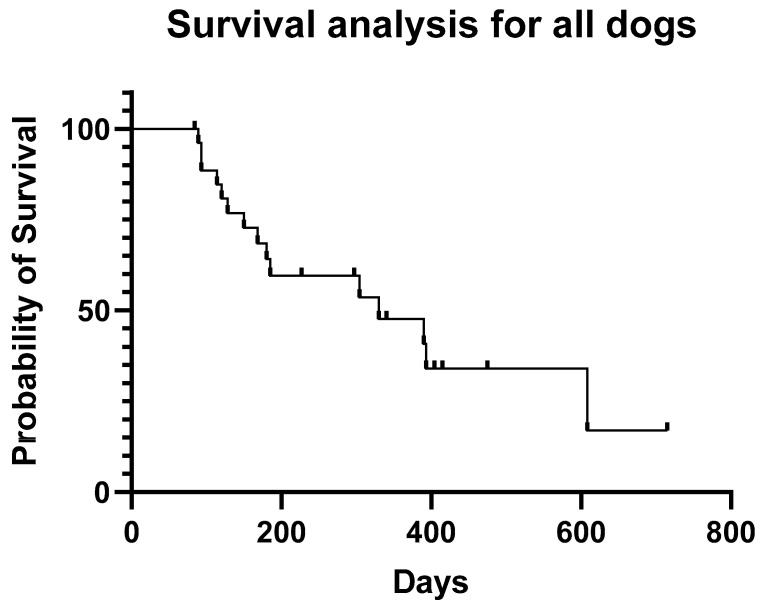
Kaplan–Meyer survival analysis for all patients. Dots represent censored cases.

**Figure 2 animals-12-02340-f002:**
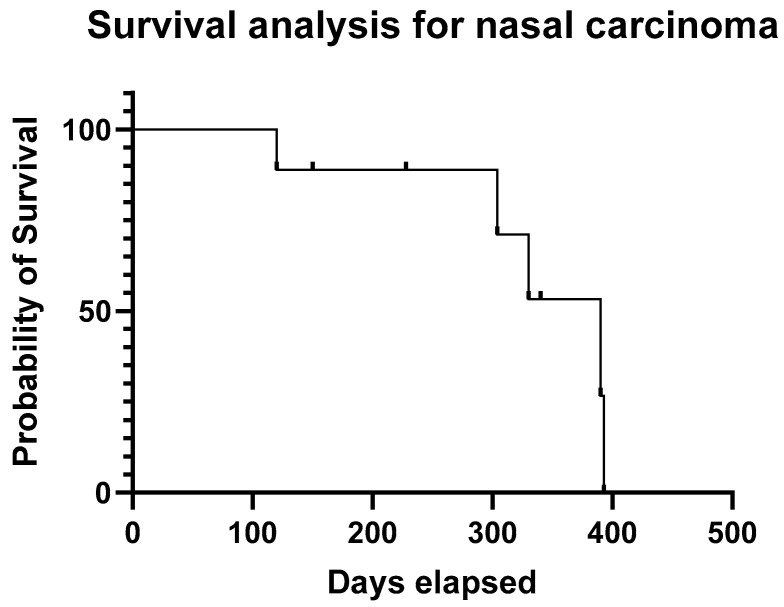
Kaplan–Meyer progression-free survival analysis for all patients with sino-nasal carcinoma. Dots represent censored cases.

**Figure 3 animals-12-02340-f003:**
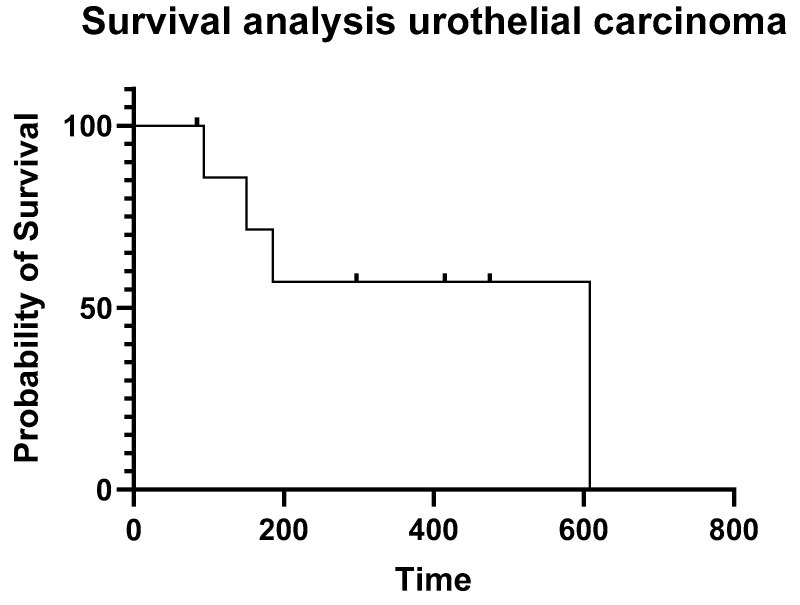
Kaplan–Meyer progression-free survival analysis for all patients with urothelial carcinoma. Dots represent censored cases.

**Table 1 animals-12-02340-t001:** Patient’s data.

Patient	Breed Sex, Age	Diagnosis	Clinical Response	Objective Response	ST	PFS
1	Schnauzer MN-11YO	Nasal carcinoma	Yes	SD	393 d	270 d
2	Shiba Inu MN-11YO	Nasal carcinoma	Yes	NE	304 d	NE
3	Chihuahua MN-11YO	Nasal carcinoma	Yes	NE	340 d ^c^	NE
4	Bichon Frise FS-13 YO	Nasal carcinoma	No	NE	NE	NE
5	Golden retriever FS-13YO	Nasal carcinoma	Yes	PR	390 d	310 d
6	Toy poodle MN-14YO	Nasal carcinoma	Yes	PR	120 d ^c^	105 d
7	Toy poodle FS-12YO	Maxillary SCC	Yes	SD	168 d	45 d
8	Mini pinscher MN-16YO	Maxillary SCC	Yes	PR	404 d ^a^	387 d
9	Corgi MN-8YO	lingual SCC	No	PD	93 d	21 d
10	Toy poodle FN-8YO	Nasal carcinoma	Yes	NE	120 d	NE
11	Toy poodle MN-15YO	Nasal carcinoma	Yes	NE	330 d	NE
12	Golden retriever FS-12YO	Nasal carcinoma	Yes	SD	150 d ^c^	56 d
13	WHWT FN-13YO	Tonsil carcinoma	No	PD	128 d	21 d
14	Schnauzer MN-11YO	Ear canal carcinoma	Yes	PR	715 d ^a^	666 d
15	Mongrel FN-12 YO	Bladder TCC	Yes	SD	608 d	437 d
16	Dachshund MN-9YO	Cutaneous SCC	Yes	PR	180 d ^c^	105 d
17	Pekinese MN-16YO	Nasal carcinoma (SCC)	No	PD	NE	21 d
18	Sheltie FN-8YO	Urethra TCC	Yes	SD	150 d	60 d
19	Dachshund MN-11YO	Prostatic carcinoma	Yes	PD	93 d	21 d
20	Corgi MN-14YO	Pulmonary carcinoma	Yes	SD	180 d	63 d
21	Mini collie MN-11YO	Bladder TCC	Yes	SD	297d ^c^	242 d
22	Toy poodle MN-13YO	Bladder TCC	No	SD	415 d ^a^	415 d
23	Scottish perrier MN-12YO	Prostatic TCC	Yes	SD	185 d	98 d
24	Toy poodle FS-9YO	Bladder TCC	Yes	SD	475 d ^b^	352 d
25	Jack Russell FS-14YO	Pulmonary carcinoma	Yes	SD	227 d ^a^	227 d
26	Toy poodle MN-9YO	Prostatic TCC	Yes	SD	84 d ^b^	84 d
27	Corgi FS-10YO	Pulmonary carcinoma	Yes	PD	114 d	21 d
28	Toy poodle FN-6 YO	Mammary carcinoma	No	PD	NE	21 d
29	Bichon Frise, MN-12YO	Frontal sinus SCC	Yes	PR	228 d ^a^	228 d
30	Bichon Frise, MN-15YO	Maxillary SCC	Yes	SD	89 d	89 d

Data of the 30 patients treated with carboplatin and bleomycin, ST (survival time), PFS (progression free survival), MN (male neutered), FN (female neutered), SD (stable disease), PR (partial response), PD (progressive disease), SCC (squamous cell carcinoma), d (days), ^a^ alive, ^b^, lost follow up, ^c^, death of unrelated causes, NE (not evaluated).

**Table 2 animals-12-02340-t002:** Summary of gastrointestinal and hematological adverse events. Graded by VCOG-CTCAE v2., G (grade).

Neutropenia	Number of Dogs	Percentage of Dogs
G1	4	13%
G2	2	6%
G3	1	3%
G4	0	
Trombocytopenia		
G1	4	13%
G2	2	6%
G3	0	
G4	0	
Anorexia		
G1	12	40%
G2	1	3%
G3	0	
G4	0	
Vomiting		
G1	4	13%
G2	0	
G3	0	
G4	0	
Diarrhea		
G1	3	10%
G2	0	
G3	0	
G4	0	

## Data Availability

The data presented in this study are available on request from the corresponding author.

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
