# Peer review of "Retrospective Evaluation of a Combination of Carboplatin and Bleomycin for the Treatment of Canine Carcinomas"

_animals, 2022, doi:10.3390/ani12182340_

Round 1

Reviewer 1 Report

The manuscript describes a retrospective study enrolled medical records of patients treated with carboplatin and bleomycin and evaluated the clinical effect. However, authors can go far with their conclusion because they have no control group. It will be very important to have at least a group (historical control) with patients treated with carboplatin (ideal scenario: there groups – carboplatin only, bleomycin only and combination of both). However, having only the carboplatin control (probably more common to find), it will be interesting.

Bleomycin is a hydrophilic drug that presents a very low intracellular absorption. For this reason, is one of the chemotherapy agents used on electrochemotherapy. Authors did not mentioned nothing about bleomycin pharmacology and low intracellular penetration. This is very important and brings the question: how will it have an antitumor effect? Please comment on that and provide statements in manuscript introduction and discussion.

The inclusion and exclusion criteria should be clearer in the manuscript

Reviewer 2 Report

Dear authros,

 The  Aminal Journal thanks you for choosing to publish the results.

 The aim of the study was to assess the safety of the combination of bleomycin and carboplatin in canine carcinomas from December 2019 to October 2021.

The simple summary e o abstract trazem as mesma informações escritas de maneira similar, sugiro reescrever o simple summary pois traz informações detalhadas que não cabe nesse item.

Line 288. “...people(18,33–36)”, change to “...people (18,33–36)”

The study evaluates the combined therapy of bleomycin and carboplatin, but the authors emphasize the need for further studies. The study results are promising and will serve as a basis for future studies, the work being relevant for future work that will focus on studies of bleomycin and carboplatin.

Reviewer 3 Report

This paper (animals-1877900) aimed to establish a new treatment protocol for the combination therapy of carboplatin and bleomycin, which is rarely used in the veterinary field, by conducting retrospective studies on various cancer types. The article affirms that the combination of carboplatin and bleomycin is effective in a variety of cancer types and does not cause noticeable side effects, and therefore, it is applicable in the veterinary field. The authors have collected as many evaluable cases as possible and analyzed them elaborately to draw conclusions. Furthermore, the limitations of retrospective studies and the need for prospective studies are mentioned, so that an objective evaluation can be made.

In conclusion, I judge this paper to be highly complete and ready for publication in its current state. However, the following minor points need to be corrected.

What do table n.1 and n.2 in lines 134 and 183 of the text refer to? 2 are different from Table1 and 2? Please clarify.
